# Chemical Composition and Bioinsecticidal Effects of *Thymus zygis* L., *Salvia officinalis* L. and *Mentha suaveolens* Ehrh. Essential Oils on Medfly *Ceratitis capitata* and Tomato Leaf Miner *Tuta absoluta*

**DOI:** 10.3390/plants11223084

**Published:** 2022-11-14

**Authors:** Hannou Zerkani, Loubna Kharchoufa, Imane Tagnaout, Jamila Fakchich, Mohamed Bouhrim, Smail Amalich, Mohamed Addi, Christophe Hano, Natália Cruz-Martins, Rachid Bouharroud, Touria Zair

**Affiliations:** 1Chemistry of Bioactive Molecules and the Environment, Faculty of Science, University Moulay Ismail, B.P. 11201 Zitoune, Meknes 50070, Morocco; 2Faculty of Sciences, Mohammed First University, Oujda 60040, Morocco; 3Laboratory of Phytochemistry, National Agency of Medicinal and Aromatic Plants-Taounate, B.P. 159, Taounate 34173, Morocco; 4Laboratoire d’Amélioration des Productions Agricoles, Biotechnologie et Environnement, (LAPABE), Faculté des Sciences, Université Mohammed Premier, Oujda 60000, Morocco; 5Laboratoire de Biologie des Ligneux et des Grandes Cultures, INRAE USC1328, Campus Eure et Loir, Orleans University, 45067 Orleans, France; 6Faculty of Medicine, University of Porto, 4099-002 Porto, Portugal; 7Integrated Crop Production Unit, Regional Center of Agricultural Research of Agadir, National Institute of Agricultural Research (INRA), Avenue Ennasr, B.P. 415 Rabat Principale, Rabat 10090, Morocco

**Keywords:** *T. absoluta*, *C. capitata*, essential oil, chemical analysis, toxicity, LD_50_, bioinsecticides

## Abstract

The present work was aimed to study the toxicity of the essential oils of three aromatic and medicinal plants on the tomato leaf miner *Tuta absoluta* and the Medfly *Ceratitis capitata* as an alternative to conventional pesticides. We carried out a phytochemical and insecticide study of *T. zygis* L., *S. officinalis* L. and *M. suaveolens* Ehrh. essential oils (EOs) through the study of their chemical composition and their toxicity on *C. capitata* adults and *T. absoluta* larvae. The extraction of the EOs by hydrodistillation showed yields of 3.87 ± 0.03, 4.09 ± 0.23 and 4.35 ± 0.11 for *T. zygis*, *S. officinalis* L. and *M. suaveolens* Ehrh., respectively. The identification of the chemical composition of the EOs by GC/MS showed that oxygenated monoterpenes constituted the most abundant group for all the extracted EOs. The major compounds were rather diversified depending on plant species. In fact, the *S. officinalis* L. EO mainly contained trans-thujone (21.80 %), the *M. suaveolens* Ehrh. EO mainly contained piperitenone oxide (71.19%), and carvacrol (61.60%) was the main component of the *T. zygis* L. EO. An insecticidal effect was observed for the three studied EOs on *C. capitata* adults and *T. absoluta* larvae. The observed LD_50_ values were 0.80 µL/mL and 11.04 µL/mL for *M. suaveolens* and *S. officinalis*, respectively, on *T. absoluta* larvae. For *C. capitata* adults, the obtained LD_50_ values were 0.9 µL/mL and 11.78 µL/mL for *M. suaveolens* and *T. zygis*, respectively. The presented findings could contribute to the development of biopesticides for plants as a component of integrated pest management strategies in citrus and tomato crops.

## 1. Introduction

Medicinal plants are a very valuable source of derivatives such as essential oils and new drugs, since they contain bioactive molecules possessing pharmacological properties, particularly lamiaceae species, which are mainly used for musculoskeletal, skin, circulatory and digestive disorders in traditional Chinese medicine [1]. Ethnopharmacological work has highlighted the fact that medicinal plants are used in various cultures and regions for shelter, food, medicines, and religious customs [2], in addition to the active molecules of medicinal plants used in the pharmaceutical industry. The industrial rational for the use of these plants lays the foundations for innovations in the development of new products for medical, veterinary, and cosmetic applications [3]. Among the active products are essential oils that show remarkable effects in the health field. The biological activities of EOs are the subject of a large number of publications at the international level in the pharmaceutical field. Citrus and tomato plants are the main crops in Morocco that cover the needs of the local market and generate a surplus intended for export. The potential markets of Morocco are the European Union, USA and Russia. However, these crops are threatened by a wide range of pests that affect yield and quality, the most important of which are the Mediterranean fruit fly (also called the fruit fly), *Ceratitis capitata* (Wiedemann, 1829), and the tomato leaf miner, *Tuta absoluta* (Meyrick, 1917). *T. absoluta* is an invasive pest that is difficult to control. Pesticide application is widespread and remains the main method of control, especially in open-field cropping systems. Accordingly, resistance to many chemical classes of insecticides has been described in both South America and Europe [4]. Indeed, a number of investigations have found that plants with EOs have powerful insecticidal effects [5,6,7,8,9,10]. *T. zygis* L., *S. officinalis* L. and *M. suaveolens* (Mentha genus) are species of the family *Lamiaceae*. Species of this family are known for their richness in essential oils [11]. The local name of *T. zygis* L., from the genus Thymus, is “Ziitra” or “adouchne”. *M. suaveolens* Ehrh. belongs to the genus Mentha, which includes all mints known by Moroccan locals as “Mersita” or “Timijja”. *S. officinalis* L. is a species from the genus Salvia, and its vernacular name is Salmia. Furthermore, many chemical investigations of *S. officinalis* oil have indicated that several of its compounds can be effective against mosquitoes, such as 1,8-cineole [12]. The insecticidal potential against mosquitoes has also been reported by a number of studies for *T. zygis* and *M. suaveolens* Ehrh [13,14]. These cited studies explained their choice of these three species. The present work falls within the framework of the valorization of the biological diversity of aromatic and medicinal herbs of the Middle Atlas region in Morocco through the research of bioactive substances, partly substitutable for synthetic pesticides. It was aimed to determine the chemical composition of the essential oils of *T. zygis* L., *S. officinalis* L., and *M. suaveolens* Ehrh. and to evaluate their insecticidal effectiveness against *Ceratitis capitata* adults and *Tuta absoluta* larvae.

## 2. Results and Discussion

### 2.1. The Yield of Essential Oils

After the extraction of the essential oils from the studied plants, the obtained yields showed that these species had quite significant contents of EOs (Table 1). The average EO yield obtained from the aerial part of *T. zygis* L. collected in the El-Hammam Khenifra area at full bloom was 3.87 ± 0.03%. This yield was greater than that obtained from the same species collected in the Timhdit Moyen Atlas area of Morocco (2.37 ± 004% (*v*/*w*)) [15]. A study by Yacoubi et al., 2014 found that the EO yields of *T. Zygis* L., in full bloom, harvested in the region of Timhdit, Elhajeb, and Azrou in Morocco were, respectively, around 5.98%, 4.16%, and 4.07% [16]. These yields were high compared with our results. According to Goncalves et al. [17], yields in Portugal varied from 1.0 to 1.5%. Both Moroccan and Portuguese yields of *T. zygis* L. EOs have been shown to be quite variable. The EO yield of *T. zygis* L. was found to be higher than the yields of other Moroccan thyme species, such as *T. Blecherianus* and *T. Riatarum* collected from Taza, with yields of 2 and 0.5%, respectively [18]. In addition, *T. satureioides* from the central High Atlas showed a yield of 2.74% [19]. The extraction of *S. officinalis* L. led to an EO yield of 4.09 ± 0.23%. Our results showed similarities with those of Khiya et al. [20], who found an EO yield of 4.13% from the Khenifra region of Morocco, which was higher than the 1.92% yield obtained from Agadir region [21]. The EO content of *S. officinalis* L. from Morocco was found to be significantly higher than those found in Italy (0.70%) [22] and Iran (0.90%) [23]. The yield of the *M. suaveolens* Ehrh. EO from Khenifra in this study was 4.35 ± 0.11% (mL/100 g dry matter). This yield was similar to that of *M. suaveolens* Ehrh. Previously harvested in the same region [24]. The EO yields for plant species harvested in the Loukkos and the Middle Atlas regions of Morocco were, respectively, found to be around 1.6% and 0.7% [25]. Furthermore, a content of 0.53% was recorded for a sample taken in the Boulmane area of Morocco [26].

### 2.2. Chemical Composition of T. zygis L., S. officinalis L. and M. suaveolens Ehrh. Essential Oils

The analysis of the EOs of *M. suaveolens* Ehrh., *S. officinalis* L. and *T. zygis* L. via gas chromatography provided spectra showing that the dominant compounds had retention times of 22.15, 20.35 and 10.81 min, respectively. Other compounds at lower percentages were also highlighted by low amplitude peaks (Figure 1). The results indicated the presence of 24 compounds comprising a total of 99.92% of the chemical composition of the EO of *T. zygis* L., 33 compounds comprising a total of 99.89% of the EO of *S. officinalis* L., and 25 compounds comprising a total of 99.79% of the total chemical composition of the EO of *M. suaveolens* Ehrh (Table 2).

The results of the chemical composition analysis revealed that oxygenated monoterpenes constituted the most frequent category of all chemical components identified in the EOs of *T. zygis* L., *S. officinalis* L., and *M. suaveolens* Ehrh, with percentages of 68.62%; 55.62% and 91.66%, respectively. The EO of *T. zygis* L. mainly consisted of carvacrol (61.60%), o-cymene (16.90%), ɤ-terpinene (7.23%) and linalool (3.14%) (Figure 2). Our results confirmed those found by Cherrat et al., 2018, who discovered the same major components with a variation in percentages for *T. zygis* L. collected from the Timhdit area [27]. However, the study conducted by Yacoubi et al., 2014 on three samples of *T. zygis* L. from the Middle Atlas found that the three samples were mostly composed of carvacrol (16.07 to 74.33%), p-cymene (6.97 to 40.26%), thymol (1.47 to 32.46%), and -terpinene (2.68 to 22%) [16]. Furthermore, the EO of *T. zygis* L. from Meknes was shown to be dominated by thymol (44.17%), p-cymene (15.45%), carvacrol (13.48%) and *γ*-terpinene (11.30%) [28]. In addition, a variation of the chemical components was observed for the EO of *T. zygis* L. collected from the Ozoud-azilal (High Atlas-Morocco) area, of which thymol (34.07%) and borneol (25.28%) were the two main constituents [29]. The *T. zygis* L. EO from Morocco is therefore characterized by an abundance of thymol or carvacrol (isomeric components). A study carried out in Portugal by Goncalves et al. showed that the chemical compositions of *T. zygis* L. species from four regions showed significant differences in terms of the main constituents [30]; in general, these major compounds were carvacrol (25%), thymol (23.8%), geranyl acetate/geraniol (20.8% and 19.8%) and linalool (30%). In the region of Paca in France, the EO of *T. zygis* L. was shown to be characterized by a high thymol content (84.9%) and less significant proportions of linalool (1.4%), p-cymene (9.7) and terpinene (2.3%) [31]. Generally, thymol and carvacrol are the main compounds found in most EOs of *T. zygis* L. The most notable variations concern compounds with lower percentages such as linalool, cymene, terpinene and borneol. However, non-aromatic terpenes may also be present as major constituents. Regarding the EO of *S. officinalis* L., we found that it was dominated by six compounds whose proportions were higher than 6%. Among these compounds, we observed three oxygenated monoterpenes, trans-thujone (21.80%), camphor (15.33%), and 1,8-cineole (11.46%); a hydrogenated monoterpene α-pinene (7.46%); and sesquiterpenes such as caryophyllene <(E)-> (6.76%) and viridiflorol (6.20%). The chemical composition observed in the current study showed similarities with the chemical profile of *S. officinalis* L. from the Khenifra region studied by Khiya et al. This profile was characterized by an abundance of thujone (trans) (17.74%), 1,8-cineole (12.63%) and camphor (12.24%) [20]. In addition, the chemical composition of *S. officinalis* L. from the Agadir region was found to be dominated by thujone (trans) (24.05%), camphor (17.16%) and 1,8-cineole (16.77%) [21]. Bouajaj et al. noted an abundance of thujone (trans) (29.84%), camphor (9.14%), 1,8-cineole (16.82%), and viridiflorol (9.92%) in a studied EO of the Ourika-Marrakech, Morocco region [32]. According to this work, the chemical composition of the EO of *S*. *officinalis* L. is quite preserved in Morocco. Regardless of the studied plant material’s region of origin in Morocco, the majority of constituents are the same (trans-thujone, camphor, and cineole), with some differences in terms of proportions. However, we observed some differences in the chemical compositions of the EOs of our sample and those in the literature from elsewhere in the world. Vergine et al. presented an EO dominated by longifolene (16.09), α-thujone (14.41), 1,8-cineole (13.93%), humulene (12.39%) and camphor (9.16%) [22]. Hayouni et al. identified 1,8-cineole (33.27%), thujone (trans) (18.40%), and thujone (cis) (13.45%) in the EO of *S. officinalis* from Tunisia [33]. In this study, the EO of *M*. *suaveolens* Ehrh. was found to be characterized by an abundance of piperitenone oxide (74.57%) and borneol (9.57%). Several other compounds were identified for the three species but in relatively small proportions (e.g., caryophyllene oxide and spathulenol). In other research conducted in Morocco, piperitenone oxide was also found to be a major compound of the EO of *M*. *suaveolens* Ehrh. collected in the Meknes region (Morocco) with a percentage of 34% [34]. The research conducted by Zekri et al., 2013 on two EOs of *M. suaveolens* Ehrh. obtained from two different localities showed that piperitenone oxide predominated, reaching 74.69% in the sample from the Azrou region and 81.67% in the sample from Mrirt [35]. Similarly, piperitenone oxide was shown to predominate in samples from Loukkos and Middle Atlas (Morocco), with percentages of 53.12% and 54.51%, respectively [25]. In contrast, the chemical compositions of EOs from Beni Mellal and Boulmane were completely different from those of our sample because of their proportions of pulegone (85.5%) and menthol (40.50%), respectively [26,32]. In Uruguay and Greece, studies have shown a predominance of piperitone oxide that reached 80.8% [36] and 62.4% [37], respectively. On the other hand, in Beheira (Egypt), the studied EO was dominated by linalool (35.32%) and p-menth-1-en-8-ol (11.08%) [38]. Different studies have shown that the chemical composition of the EO of *M*. *suaveolens* Ehrh. somewhat varies, but the essential components are piperitenone oxide, menthol, piperitenone, pulegone, and linalool.

### 2.3. Insecticidal Activity

#### 2.3.1. Toxicity of Essential Oils

After exposing *C. capitata* adults and *T. absoluta* larvae to various concentrations of the essential oils for 24 h and 48 h, mortality rates were found to vary with concentration (Figure 3A,B), and the results showed very high significant effects (*p* = 0.000) for the three tested EOs at 24 and 48 h after treatment. For the *T. zygis* L. EO, we found that after 24 h of exposure, a dose of 50 µL/mL led to a 100% mortality rate for *C. capitata* adults. After 48 h of exposure, the adults of *C. capitata* were fully controlled at a concentration of 25 µL/mL. For *T. absoluta* larvae, a concentration of 6.25 µL/mL of the *T. zygis* L. EO led to 90.36% and 100% mortality after 24 h and 48 h of exposure, respectively. The exposure of the *S. officinalis* L. EO led to the total mortality of *C. capitata* adults at a dose of 25 µL/mL after 24 h of exposure. For *T. absoluta* larvae, a dose of 25 µL/mL of the *S. officinalis* L. EO led to 96.36% and 100% mortality rates after 24 h and 48 h of exposure, respectively. 

#### 2.3.2. Determination of LD_50_ of the Studied Essential Oils 

A comparison of the toxicity, in terms of LD_50_, of the extracts of the three studied plants is presented in Table 3. The analysis of the variance of LD_50_ (lethal dose for 50% of the population) revealed significant variations between the three essential oils. The LD_50_ of the *T. zygis* L. EO, after 24 h and 48 h of exposure against *C. capitata* was, respectively, 11.78 µL/mL and 7 µL/mL. The LD_50_ of this EO against *T. absoluta* larvae was, respectively, 6.18 µL/mL and 3.76 µL/mL after 24 h and 48 h of exposure. Indeed, the insecticidal efficacy exerted by the EO of *T. zygis* L. against *T. absoluta* larvae may have been due to the richness of this EO in carvacrol (61.60%). These findings are compatible with those of the study of Bouayad Alam et al., 2017, who found that the EO of *T. capitatus* (whose percentage of carvacrol is 69.6%) had a very interesting toxic effect on the larval stages of *T. absoluta* at low concentrations [39]. The lethal dose of LD_50_ of the *S. officinalis* L. EO to *C. capitata* adults was found to be about 6.56 µL/mL at 24 h of exposure and 3.44 µL/mL at 48 h of contact. The LD_50_ (by the same species) of *T. absoluta* larvae was 11.04 µL/mL after 24 h of exposure and 8.38 µL/mL after 48 h. The EO of *S. officinalis* L. demonstrated a significant insecticidal effect. This EO is rich in trans-thujone (21.80%), camphor (15.33%), and 1,8-cineole (11.46%). Xie et al. found that α-thujone, β-thujone, and 1,8-cineole tested alone showed a strong toxicity toward *Solenopsis invicta* [40]. In our study, the LD_50_ of the *M. suaveolens* Ehrh. EO towards *T. absoluta* larvae after 24 h and 48 h of exposure was around 1.13 µL/mL and 0.80 µL/mL, respectively. On the other hand, the EO of *M. suaveolens* Ehrh. showed an LD_50_ of 2.3 µL/mL towards *C. Capitata* after 24 h of exposure and 0.9 µL/mL after 48 h of exposure. The low calculated LD_50_ confirmed the high degree of toxicity of this EO. Insecticidal test results showed that the *M. suaveolens* Ehrh. EO was highly toxic against *T. absoluta* larvae and *C. capitata* adults—even the g value obtained by probit analysis was higher than 0.5. Numerous investigations of the toxicity of the *M. suaveolens* Ehrh. EO against several insects have been performed. Indeed, Boughdad et al. noticed that this EO, rich in piperitone oxide (86.2%), showed a significant efficiency against *C. Maculatus* [34]. The toxicity of the tested oils could be attributed to the high level of piperitone oxide (71.19%). According to previous studies, this oxygenated monoterpene has a high toxicity against harmful organisms [35,37,41,42]. In general, results have demonstrated that the type, quantity, and exposure period of oils, as well as the insect used, have impacts on the insecticidal effect of EOs. The tested EOs demonstrated clear insecticide action on *T. absoluta* larvae and *C. capitata* adults. This important activity can be attributed to a synergy between major and minor EO constituents and frequently leads to a better insecticide ability.

## 3. Materials and Methods

### 3.1. Plant Material

*T. zygis* L. (Figure 4A), *S. officinalis* L. (Figure 4B) and *M. suaveolens* Ehrh. (Figure 4C) were collected in the rural commune of El Hammam (Khenifra-Middle Atlas-Morocco) (Figure 5) in full bloom. Then, they were dried in the shade for about ten days. The plant species were botanically identified at the floristic laboratory of the Scientific Institute of Rabat. The voucher specimens of the 3 species were preserved in an herbarium at INRA-Agadir and labelled as TZ-2017, SO-2017, and MS-2017, respectively.

### 3.2. Essential Oil Extraction

Using Clevenger equipment, the hydrodistillation process was used to extract the EOs. For each test, 100 g of raw material was processed, and about three hours were spent in the extraction process. Per sample, at least three iterations were performed. The level of humidity of the various samples was calculated in order to represent the yield of the EOs (*v*/*w*) (volume in mL) per 100 g of the dry matter. Then, the EOs were dehydrated on anhydrous sodium sulfate and stored at 4 °C away from light until needed [43].

### 3.3. Analysis of the Chemical Composition of the Essential Oils

The chromatographic analysis of the EOs was conducted with a gas chromatograph of the Thermo Electron type (Trace GC Ultra) linked to a Thermo Electron Trace MS system (Thermo Electron: Trace GC Ultra; Polaris Q MS). An intensity of 70 eV was used for the fragmentation. The chromatograph was equipped with a column of the DB-5 type (5% phenyl-methyl-siloxane) (30 m × 0.25 mm × 0.25 μm film thickness) and a flame ionization detector (DIF) powered by a H2/Air gas mixture. The temperature of the column was programmed to increase at 4 °C/min from 50 to 200 °C over 5 min. The injection mode was split (leak report: 1/70, flow mL/min), and the used vector gas was nitrogen at a flow rate of 1 mL/min. The chemical composition of the EOs was determined based on a comparison of their indices of calculated Kovats (KI) with those of Adams, 2007 [44] Kovats indices can be used to compare the retention time of any product with that of a linear alkane with the same number of carbons. These indices are determined by injecting a mixture of alkanes (standard C7-C40) under the same operating conditions.

They are calculated used the following formula:IK = [(TRx − TR_n)/(TR_(n + 1) − TR_n) + n] × 100
where TRx is the retention time of the solute x, and TRn and TR (n + 1), which frame the peak of the solute, are the retention times of the linear alkanes at n and n + 1 of the carbon atoms, respectively. The retention index of a compound to be analyzed is independent of the flow rate, the length of the column, and the quantity, but it depends on the stationary phase and the temperature (T).

### 3.4. Insecticidal Activity

#### 3.4.1. Insects Material

To investigate the insecticidal effects of the essential oils of *T. zygis* L., *S. officinalis* L., and *M. suaveolens* Ehrh, two insects—adults of *C. capitata* and larvae of *T. absoluta*—were used (L2).

The development of *Ceratitis* essentially depends on thermal conditions. Mediterranean fly breeding is relatively simple. Collected fallen argan fruits (with bites and small stains on their surface; they contain eggs) are placed in a 30 L box containing sand. After 3 to 5 days of incubation, the hatching starts, and the larvae from the eggs complete their development in the fruit pulp. When they have finished developing, they leave the fruit to pupate in the ground; the pupa is also the fruit fly’s resistance stage. The complete cycle of this species is completed in 15 to 17 days, depending on the laboratory temperature. Afterwards, the flies begin to lay eggs through muslin.

Here, two automated greenhouses were used, the first to produce healthy, untreated tomato leaves and the second to rear *T. absoluta*. The healthy tomato leaves and the larvae of *T. absoluta* used in the bioassays were collected from the experimental greenhouses of the Melk Zhar-Belfaa (DEMZ)-INRA-Agadir-Morocco experimental farm, which are characterized by a relative humidity of 65% ± 5% and a temperature of 26 ± 2 °C. The two greenhouses did not undergo any pesticide treatment in order to not influence the effects of the EOs on *T. absoluta*.

#### 3.4.2. Concentrations Preparation

Doses were prepared for each plant by dissolving pure essential oils in di-methyl sulfoxide. Using this initial concentration (100 L/mL), we prepared a range of concentrations of 50, 25, 12.5, 6.25, 3.125, and 1.562 L/mL. Di-methyl sulfoxide alone was used as a control.

#### 3.4.3. Biological Trials

Larvae of *T. absoluta*

The “leaf-dip bioassay” developed by Galdino et al., 2011 was the technique used in our tests with few modifications [45]. This method consists of dipping 4 tomato leaflets in each concentration for 10 s before laying them out on blotting paper for 5 min to air-dry. Afterward, every treated leaflet was put into a box. Five larvae of *T. absoluta* were carefully placed in each box using a fine brush. Then, these boxes were maintained in a laboratory environment with 26 ± 2 °C of temperature, 60–70% humidity, and a 16–8 photoperiod (light/obscurity). The mortality rates were calculated after 24 h and 48 h. Under the same conditions, the bioassay was carried out three times for each extract. Therefore, each concentration required a total of 12 repetitions. For each bioassay, a control was prepared under the same conditions.

Adult of *C. capitata*

A temperature of 28 °C was used in the laboratory to conduct the bioassay. Circular sponges with a diameter of 5 cm and a thickness of 0.5 cm were soaked in the various concentrations and placed in 400 mL boxes, and then 10 adult fruit flies were fed into each box with a pump. The boxes’ coverings were carved into a circle in the center and hermetically sealed with mosquito nets to ensure ventilation and the better control of the observation while also preventing the escape of *C. capitata*. To evaluate the impact of the EOs, a control was conducted. For each essential oil, the bioassay was conducted three times under the same conditions. Consequently, there were 12 repetitions in total for each concentration and control.

### 3.5. Data Analysis

The percentages of *C. capitata* adult and *T. absoluta* mortality were determined for the 3 tested EOs. The efficacy of the EOs on the mortality rates of these 2 pests was subjected to a one-way analysis of variance test (ANOVA) using the IBM SPSS general linear model (GLM) procedure, and then the Student–Newman–Keuls test was conducted for mean comparisons. A probit analysis of the concentration-dependent mortality data was conducted using POLO-PC (LeOra Software, 1987). The following parameters were calculated: LD50, fiducial limits, standard error, and slope of regression.

## 4. Conclusions

At the end of this study, the yields of the EOs of the studied plants varied between 3.87 ± 0.03 for *T. zygis* L. and 4.35% ± 0.11 for *M. suaveolens* Ehrh. Carvacrol (61.60%) and o-cymene (16.90%) constituted the major compounds of the EO of *T. zygis* L., while the EO of *S. officinalis* L. was dominated by trans-thujone (21.80%), camphor (15.33%) and 1,8-cineole (11.46%). The EO of *M. suaveolens* Ehrh. was characterized by an abundance of piperitone oxide (74.57%) and borneol (9.57%). The tested EOs demonstrated clear insecticide activity on *Tuta absoluta* larvae and *Ceratitis capitata* adults. For the *T. zygis* L. EO, we found that after 24 h of exposure, a dose of 50 µL/mL led to a 100% mortality rate against *C. capitata* adults. A concentration of 6.25 µL/mL of the *T. zygis* L. EO led to a 90.36% mortality of *T. absoluta* larvae. At a dose of 12.5 µL/mL, the *S. officinalis* L. EO destroyed the adults of *C. Capitata* after 48 h of exposure. For the EO of *M. suaveolens* Ehrh., we note that regardless of the tested species (*C. capitata* and/or *T. absoluta*) and the duration of exposure, we obtained mortality of 100% at a concentration of 3.125 µL/mL. This significant effect can be due to a synergy between major constituents and other minority EO compounds that often leads to better outcomes. The obtained results lead us to look far and to open vistas for the valorization and exploitation of these plants and their chemical compounds. The current findings could contribute to the development of biopesticides for plants as a component of integrated pest management strategies in citrus and tomato crops.

## Figures and Tables

**Figure 1 plants-11-03084-f001:**
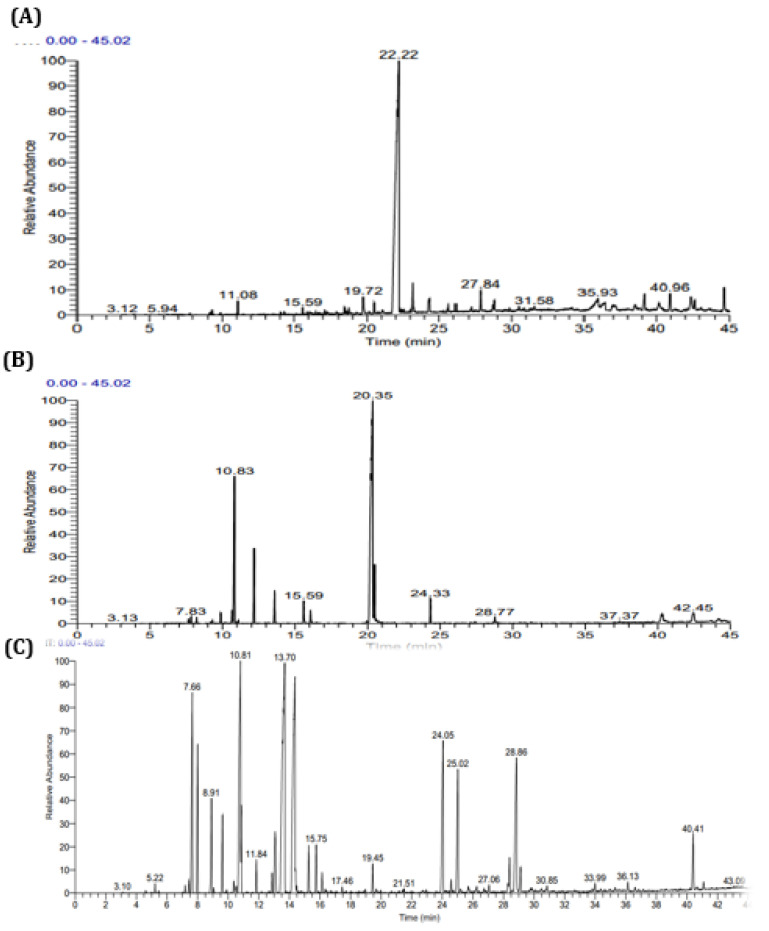
Chromatogram of *M. suaveolens* Ehrh. (**A**), *T. zygis* L. (**B**), and *S. officinalis* L. (**C**) EOs.

**Figure 2 plants-11-03084-f002:**
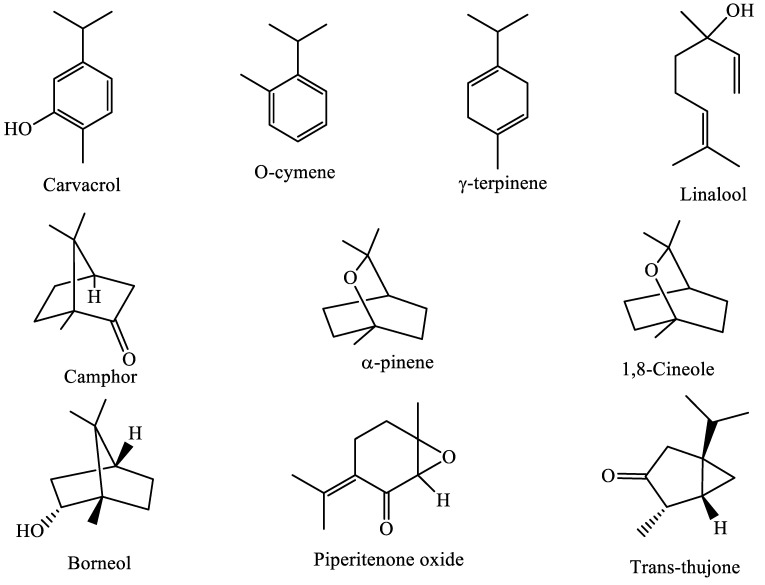
Chemical structures of the most abundant component of the three studied OEs.

**Figure 3 plants-11-03084-f003:**
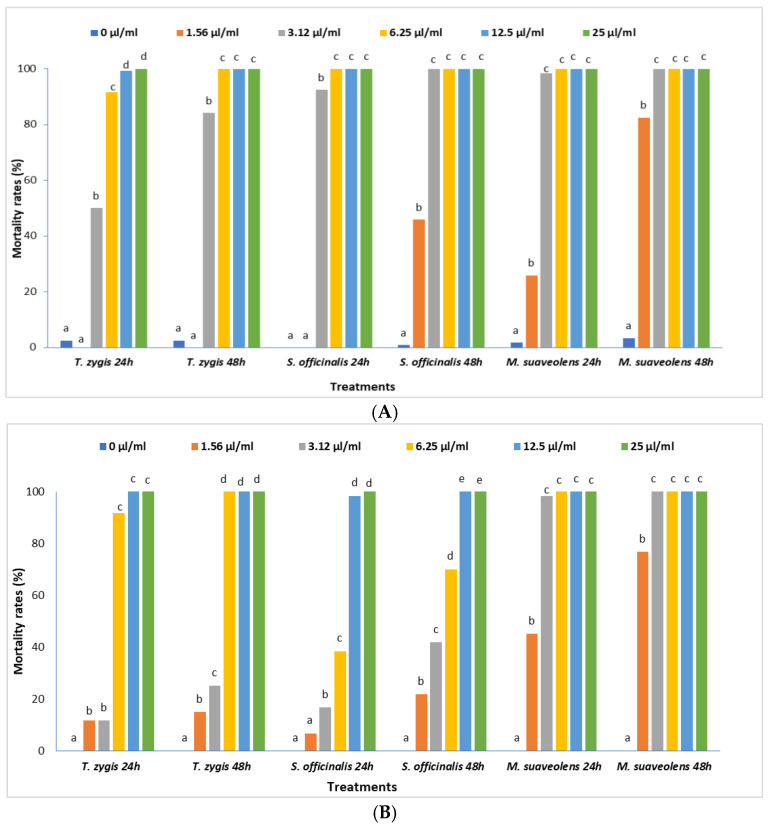
Response of *C. capitata* adults (**A**) and *T. absoluta* larvae (**B**) to different concentrations of the tested EOs. By treatment, the rates followed by the same letters were not statistically different at *p* < 0.05% according to the Newman–Keuls test.

**Figure 4 plants-11-03084-f004:**
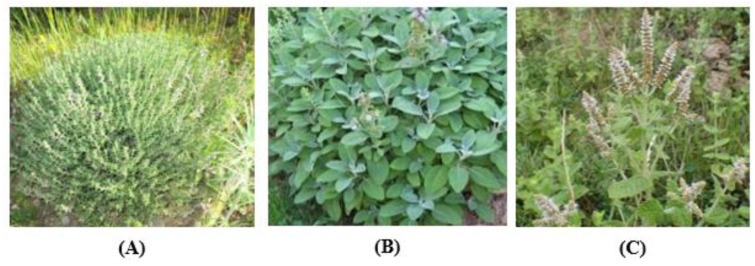
*T. zygis* L. (**A**), *S. officinalis* L. (**B**) and *M. suaveolens* Ehrh. (**C**).

**Figure 5 plants-11-03084-f005:**
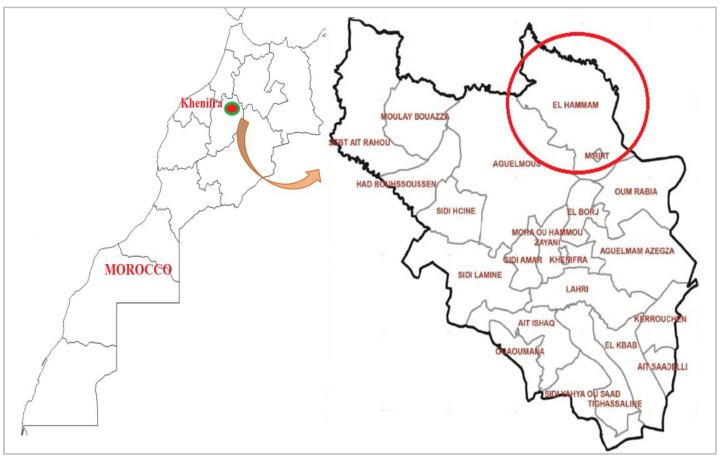
Collection site of the studied plants.

**Table 1 plants-11-03084-t001:** Yields of essential oils extracted from the studied plants.

Plant	*T. zygis* L.	*S. officinalis* L.	*M. suaveolens*
**Yield (%*v*/*w*)**	3.87 ± 0.03	4.09 ± 0.23	4.35 ± 0.11

**Table 2 plants-11-03084-t002:** Chemical composition of the EOs of *T. zygis* L., *S. officinalis* L. and *M. suaveolens* Ehrh.

RT	Constituents	(KI)	Formula	Percentage (%)		
				*T. zygis L.*	*S. officinalis L.*	*M. suaveolens* Ehrh.
5.22	Salvene <(Z)->	856	C_9_H_16_	-	0.16	-
7.19	tricyclene	926	C_10_H_16_	-	0.17	-
7.67	Thujene <α->	930	C_10_H_16_	0.33	0.34	-
7.83	Pinene <α->	939	C_10_H_16_	0.50	7.46	-
8.22	Camphene	954	C_10_H_16_	0.46	4.40	-
9.16	Pinene <β->	979	C_10_H_16_	0.11	2.69	-
9.31	Octen-3-ol <1->	979	C_8_H_16_O	0.27	-	0.24
9.89	Myrcene	990	C_10_H_16_	0.90	2.14	-
10.19	Phellandrene <α->	1002	C_10_H_16_	0.07	-	-
10.67	Terpinene <α->	1017	C_10_H_16_	1.02	0.33	-
10.81	Cineole <1.8>	1031	C_10_H_18_O	-	11.46	-
10.83	Cymene <ο->	1026	C_10_H_14_	16.90	0.27	-
11.01	Phellandrene <β->	1029	C_10_H_16_	0.14	-	-
11.08	Limonene	1029	C_10_H_16_	0.27	1.20	0.77
12.18	Terpinene <γ->	1059	C_10_H_16_	7.23	0.81	-
12.88	Terpinolene	1088	C_10_H_16_	0.06	0.51	-
13.07	Thujone (cis)	1102	C_10_H_16_O	-	2.76	-
13.6	Linalool	1096	C_10_H_18_O	3.14	-	-
13.70	Thujone (trans)	1114	C_10_H_16_O	-	21.80	-
14.37	Camphor	1146	C_10_H_16_O	-	15.33	-
15.27	Isoborneol	1160	C_10_H_18_O	-	1.40	-
15.59	Borneol	1169	C_10_H_18_O	2.18	-	9.57
15.75	Terpinen-4-ol	1177	C_10_H_18_O	1.24	1.54	0.24
16.14	Terpineol <α->	1188	C_10_H_18_O	-	0.52	-
18.43	Carvone oxide <cis>	1263	C_10_H_14_O_2_	-	-	0.47
18.73	Carvone	1243	C_10_H_14_O	-	-	0.38
19.45	Isobornyl acetate	1285	C_12_H_20_O_2_	-	0.81	-
19.72	Carvotanacetone <6-hydroxy->	1310	C_10_ H_16_ O_2_	-	-	1.55
19.93	Thymol	1290	C_10_H_14_O	0.19	-	-
20.35	Carvacrol	1299	C_10_H_14_O	61.6	-	0.75
22.22	Piperitenone oxide	1368	C_10_H_14_O_2_	-	-	74.57
22.37	Jasmone <dihydro->	1380	C_11_H_18_O	-	-	0.43
22.55	2,4-Dimethyl-1,3-cyclopentanedione	1390	C_7_H_10_O_2_	-	-	0.16
23.04	Jasmone <(Z)->	1392	C_11_ H_16_ O	-	-	0.20
24.05	Caryophyllene <(E)->	1419	C_15_H_24_	2.42	6.76	-
23.19	Nepetalactone <4aα,7α,7aβ->	1387	C_10_H_14_O_2_	-	-	3.10
24.35	Caryophyllene <(Z)->	1408	C_15_H_24_	-	-	0.84
24.58	Aromadendrene	1441	C_15_H_24_	-	0.39	-
25.02	Humulene <α->	1454	C_15_H_24_	0.08	4.93	-
25.66	Farnesene <(E)-β->	1456	C_15_H_24_	-	-	0.44
25.69	Muurolene (ɣ)	1479	C_15_H_24_	-	0.24	-
26.14	Germacrene D	1481	C_15_H_24_	-	-	0.94
27.06	Cadinene <δ->	1523	C_15_H_24_	0.15	0.24	1.51
27.2	Calamenene <trans->	1522	C_15_H_22_	-	-	0.26
28.28	Spathulenol	1578	C_15_H_24_O	-	-	0.43
28.40	Caryophyllene oxide	1583	C_15_H_24_O	0.40	1.16	0.70
28.73	Globulol	1590	C_15_H_26_O	-	-	0.63
28.86	Viridiflorol	1592	C_15_H_26_O	-	6.20	-
29.12	Humulene epoxide II	1608	C_15_H_24_O	-	0.79	-
29.44	Cubenol <1,10-di-epi->	1619	C_15_H_26_O	-	-	0.76
30.51	Cadinol <α->	1654	C_15_H_26_O	-	-	0.30
31.27	Germacra-4(15),5,10(14)-trien-1-α-ol	1686	C_15_H_24_O	0.11	0.31	0.23
33.99	Costol (β)	1767	C_15_H_24_O	-	0.25	-
36.13	Isovalencenol (E)	1793	C_15_H_24_O	-	0.29	-
36.26	Eremophilone <8-hydroxy->	1847	C_15_ H_22_ O_2_	-	-	0.32
37.37	Farnecyl acetone -<5E, 9Z>	1887	C_8_H_30_O	0.15	-	-
40.41	Manool (13-epi)	2060	C_20_H_34_O	-	2,0	-
41.09	Himachalene <α->	1451	C_15_ H_24_		0.32	
	Total (%)			99.92	99.89	99.79
	Hydrogenated monoterpenes			27.99	20.48	0.77
	Oxygenated monoterpenes			68.62	55.62	91.66
	Hydrogenated sesquiterpenes			2.65	12.79	3.99
	Oxygenated sesquiterpenes			0.66	9	3.37
	Diterpenes				2	

**Table 3 plants-11-03084-t003:** LD*_50_* (µL/mL) of the essential oils of *T. zygis* L., *S. officinalis* L. and *M. suaveolens* Ehrh. against *T. absoluta* larvae and *C. capitata* adults based on log-dose probit mortality rate data.

Pests	Duration after Treatment (h)	DL_50_ (µL/mL)	Confidence Limits	Slope	Standard Error	Heterogeneity	g Value	Log (L)
	*T. zygis*							
*C. capitata*	24	11.78	11.78–14.00	5.75	0.48	0.61	0.03	−144.60
	48	7.00	7.00–11.00	6.36	0.76	0.55	0.05	−226.80
*T. absoluta*	24	6.18	3.00–6.18	3.50	0.34	1.83	0.07	−83.72
	48	3.76	2.00–3.76	3.42	0.34	1.15	0.05	−82.30
	*M. suaveolens*							
*C. capitata*	24	2.30	-	11.15	0.01	0.05	>0.5	−89.00
	48	0.90	-	9.14	0.00	0.16	>0.5	−79.05
*T. absoluta*	24	1.13	-	16.94	0.00	0.03	>0.5	−41.29
	48	0.80	-	15.33	0.03	0.12	>0.5	−32.60
	*S. officinalis*							
*C. capitata*	24	6.56	6.56–7.00	4.02	0.40	0.88	0.04	−230.20
	48	3.44	3.44–4.00	2.98	0.25	0.70	0.03	−211.90
*T. absoluta*	24	11.04	10.00–13.00	3.67	0.34	0.79	0.03	−96.91
	48	8.38	6.00–8.40	2.72	0.24	0.80	0.03	−123.80

## Data Availability

All the data supporting the findings of this study are included in this article.

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
