# Peer review of "Chemical Composition and Bioinsecticidal Effects of Thymus zygis L., Salvia officinalis L. and Mentha suaveolens Ehrh. Essential Oils on Medfly Ceratitis capitata and Tomato Leaf Miner Tuta absoluta"

_plants, 2022, doi:10.3390/plants11223084_

Round 1
Reviewer 1 Report
The manuscript by Zerkani et al. studies the phytochemical composition and insecticide activity of essential oils (EOs) extracted by T. zygis L., S. officinalis L. and M. suaveolens Ehrh. The OE's chemical composition was studied by GC-MS. Their toxicity was evaluated on C. capitata adults and T. absoluta larvae.
The overall experiment's lack of novelty and the general quality is not enough to publish on Plants, in detail:
1) The quality of the English is very scant. The manuscript is full of wordy sentences or a complete misuse of words. For instance, in line 56: " Agriculture is one of the fundamental crops in Morocco...", it is clear that Agriculture is not a crop. Is possible that the authors wanted to say that Agriculture is one of the fundamental economic sectors of Marocco. In conclusion, Extensive editing of language and style is required.
2) In the Abstract and the Introduction, the objective of the investigation should be more clearly stated. Why the authors carried out this research? Why is it significant?
2) The results are reported and described only using qualitative expressions like "varies somewhat" or "we have noticed some differences". Could be useful to report the chemical structures of the most abundant component of the OE in a Figure. The statistical analysis should be properly presented ( P value?) in Table 3 and/or in Figure 2. It is essential to report the negative controls in Figure 2. Finally, should be interesting to use a positive control ( a common insecticide) in the bioassay and compare the effect of the EO with the positive control.
3) The discussion of the results is very poor. For instance:
a) Line 230: "We can say then that the toxicity of our oil can be justified by the high level of piperitone oxide (71.19%). According to studies, this oxygenated monoterpene has high toxicity against harmful organisms". Did the literature cite reports study on the bioactivity of the pure piperitone oxide on the same organism assayed in this study? If not, probably should be necessary to assay the pure piperitone oxide at the different concentrations in C. capitata adults and T. absoluta larvae.
b) Line 235: " This important activity can be due to a synergy between major and other minority EO constituents and frequently leads to a better finding." However, the author did not discuss further this assumption or did not present any literature supporting this statement.
Thus, a proper Discussion section should be included in the manuscript.
Considering all these comments the manuscript cannot be accepted for publication on Plants in the present form.
Author Response
Dear Editors, Dear Reviewers,
Please find, in the resubmission section, our final response to the comments received from the three reviewers to paper titled: Phytochemical profile and insecticidal potential of Thymus zygis L., Salvia officinalis L. and Mentha suaveolens Ehrh. essential oils against Ceratitis capitata and Tuta absoluta.
First of all, we would like to thank the editor and the reviewers for their thoughtful comments and efforts towards improving our manuscript by taking time in reading and suggesting modifications to the paper. We highly appreciate the comments received, as they have pointed out a number of issues to be addressed, which have been very useful to amend the paper .
Thank you very much for your kind consideration of this resubmitted version of our manuscript.
Sincerely yours,
Pr. Mohamed Addi and the co-authors
Response to Reviewer 1
Thank you for your review of our paper. We have answered each of your points below.
Comment 1: The quality of the English is very scant. The manuscript is full of wordy sentences or a complete misuse of words.
Response: As recommended by the referee, we have revised the typos, grammatical errors and the English language editing of the manuscript was revised by Pr. Ismail BENBIHI who is a professor of English at high school (Guelmim, Morocco).
Comment 2: In line 56: "Agriculture is one of the fundamental crops in Morocco...", it is clear that Agriculture is not a crop. Is possible that the authors wanted to say that agriculture is one of the fundamental economic sectors of Morocco.
Response: The sentence was reviewed as: ‘Citrus and tomato are the main crops in Morocco that cover the needs of the local market and generates a surplus intended for export. The potential markets of Morocco are European union, USA and Russia’.
Comment 3: In the Abstract and the Introduction, the objective of the investigation should be more clearly stated. Why the authors carried out this research? Why is it significant?
Response: The objective of the current study was highlighted in the abstract.
Comment 4: Could be useful to report the chemical structures of the most abundant component of the OE in a Figure.
Response: As recommended by the referee, we have added the chemical structures of the most abundant component of the OE in a Figure 2.
Comment 5: The statistical analysis should be properly presented ( P value?) in Table 3 and/or in Figure 2. It is essential to report the negative controls in Figure 2. Finally, should be interesting to use a positive control (a common insecticide) in the bioassay and compare the effect of the EO with the positive control.
Response: The effect of essential oil and mortality efficiency were subjected to ANOVA and Newman & Keuls test for mean comparison using SPSS software (Figure 3). The p values obtained were p=0.000 which means very highly significant effects of 3 EO’s tested 24 and 48h after treatment. The table 3 was revised and related statistic data were inserted (confidence limits, standard errors, slopes …). Regarding negative control it is considered that concentration 0 is a negative control. We are fully agree that it is important to use a positive control, however the commercial product based on essential oils in the market to control both pests (C. capitata and T. absoluta) is unavailable.
Comment 6: The discussion of the results is very poor. For instance:
Line 230: "We can say then that the toxicity of our oil can be justified by the high level of piperitone oxide (71.19%). According to studies, this oxygenated monoterpene has high toxicity against harmful organisms". Did the literature cite reports study on the bioactivity of the pure piperitone oxide on the same organism assayed in this study? If not, probably should be necessary to assay the pure piperitone oxide at the different concentrations in C. capitata adults and T. absoluta larvae.
- Line 235: "This important activity can be due to a synergy between major and other minority EO constituents and frequently leads to a better finding." However, the author did not discuss further this assumption or did not present any literature supporting this statement.
- Thus, a proper Discussion section should be included in the manuscript
Response: Thank you for your comments. Yes we agree it will be perfect if we can test piperitone oxide. It is known that toxic activity is mainly caused by the major components of the product (plant extracts or EO). For that we have assumed as other studies cited in the reference list that the major components could be responsible for most toxicity caused to both studied insects. Indeed, the synergism of minor compounds from plant extracts and EO’s was also possible according to other studies. In the discussion section, the text was revised based on that previous studies.
Reviewer 2 Report
The manuscript "Phytochemical profile and insecticidal potential of Thymus zygis L., Salvia officinalis L. and Mentha suaveolens Ehrh. essential oils against Ceratitis capitata and Tuta absoluta" is very interesting and has good scientific merits
but some points required more attention by the authors
Among these points:
1- The title is misleading usually phytochemical profile is a general and broad term that should be used carefully. I would rather recommend changing it to something that is more specific to volatile components
2- In the Abstract the general intro is not required therefore the first 3 line could be deleted in the abstract
3- In the introduction part: the authors should highlight what was done on the plants of interest rather than giving general facts about the use of medicinal plants
4- In the results part I would recommend changing it into results and discussion because comparison of the the results with others is related to discussion and the authors should explain more the similarity and differences between their results and others
5- In table 1 The authors should mention the yields is calculated based on v/w or w/w or dried weights because the raw word used in the method is vague also the authors should mention
6- In the GC/MS analysis: in Figure 1 the authors should either remove the retention times from the chromatograms and replace it with numbering matching with the table OR the retention times should be added to table 2
7- In table 2 what does IK stands for if it is Kovats Indices then it should be corrected in addition the authors reported their calculated KI only I would recommend adding the reported KI from Adam which they used
8- In The insecticide activity I would ask about the use of reference standard to compare the effects of the oils to this standard
9- In table 3 It was perfect that the authors were sticky to the use of uniform significant numbers (number of digits after the decimals) in table 3 presence of a mix is not accepted
10- The plant voucher sample numbers are totally missed
11- Reference 40 is related to Adams 2007 what references are given for other publications mention in line 273 and 274
12- Is there any possibility to statistically compare between the effects of the three oils ?
13- The manuscript should be checked by an English native speaker to remove some syntax
Author Response
Dear Editors, Dear Reviewers,
Please find, in the resubmission section, our final response to the comments received from the three reviewers to paper titled: Phytochemical profile and insecticidal potential of Thymus zygis L., Salvia officinalis L. and Mentha suaveolens Ehrh. essential oils against Ceratitis capitata and Tuta absoluta.
First of all, we would like to thank the editor and the reviewers for their thoughtful comments and efforts towards improving our manuscript by taking time in reading and suggesting modifications to the paper. We highly appreciate the comments received, as they have pointed out a number of issues to be addressed, which have been very useful to amend the paper .
Thank you very much for your kind consideration of this resubmitted version of our manuscript.
Sincerely yours,
Pr. Mohamed Addi and the co-authors
Response to Reviewer 2
Thank you for your comments. Our answers to your points are as follows
Comment 1: The title is misleading usually phytochemical profile is a general and broad term that should be used carefully. I would rather recommend changing it to something that is more specific to volatile components
Response: As suggested, the new manuscript title is currently: “Chemical composition and bioinsecticidal effects of Thymus zygis L., Salvia officinalis L. and Mentha suaveolens Ehrh. essential oils on Medfly Ceratitis capitata and tomato leafminer Tuta absoluta”.
Comment 2: In the Abstract the general intro is not required therefore the first 3 line could be deleted in the abstract.
Response: Thank you for these remarks, the general introduction was removed from abstract.
Comment 3: In the introduction part: the authors should highlight what was done on the plants of interest rather than giving general facts about the use of medicinal plants
Response: The general aspects related to medicinal and aromatic plants were removed from the introduction and introduction was focused only on the studied species.
Comment 4: In the results part I would recommend changing it into results and discussion because comparison of the the results with others is related to discussion and the authors should explain more the similarity and differences between their results and others.
Response: As you have recommended, we have replaced “results” By “results and discussion” in the manuscript.
Comment 5: In table 1 The authors should mention the yields is calculated based on v/w or w/w or dried weights because the raw word used in the method is vague also the authors should mention
Response: The yields were estimated based on V/W, it is inserted in the table 1.
Comment 6: In the GC/MS analysis: in Figure 1 the authors should either remove the retention times from the chromatograms and replace it with numbering matching with the table OR the retention times should be added to table 2
Response: As suggested by the referee, the retention time was added to table 2.
Comment 7: In table 2 what does IK stands for if it is Kovats Indices then it should be corrected in addition the authors reported their calculated KI only I would recommend adding the reported KI from Adam which they used
Response: As suggested by the referee, IK was replaced by KI.
Comment 8: In The insecticide activity I would ask about the use of reference standard to compare the effects of the oils to this standard.
Response: We fully agree that it is important to use a positive control, however the commercial product based on essential oils in the market to control both pests (C. capitata and T. absoluta) is unavailable.
Comment 9: In table 3, it was perfect that the authors were sticky to the use of uniform significant numbers (number of digits after the decimals) in table 3 presence of a mix is not accepted.
Response: The values were adjusted using only 2 decimals.
Comment 10: The plant voucher sample numbers are totally missed.
Response: “The voucher specimens of the 3 species were preserved as a herbarium at INRA-Agadir and labelled as TZ-2017, SO-2017 and MS-2017.”
Comment 11: Reference 40 is related to Adams 2007 what references are given for other publications mention in line 273 and 274.
Response: Only the Adams reference was kept here in this part.
Comment 12: Is there any possibility to statistically compare between the effects of the three oils?
Response: The percentage of C. capitata adult and T. absoluta mortality was determined for the 3 tested EO’s. The efficacy of EO’s on the mortality rates of these 2 pests were subjected to a one-way analysis of variance test (ANOVA), adopting the IBM SPSS general linear model (GLM) procedure then completed by Student-Newman-Keuls test for mean comparisons. Probit analysis of the concentration dependent mortality data was made using POLO-PC (LeOra Software, 1987). The following parameters were calculated: LD50, Fiducial Limits, Standard Error and Slope of regression.
Comment 13: The manuscript should be checked by an English native speaker to remove some syntax.
Response: We have revised the typos, grammatical errors and the English language editing of the manuscript was revised by Pr. Ismail BENBIHI who is a professor of English.
Reviewer 3 Report
This article present phytochemical profiling of the selected plants and its insecticidal activities of these plants. Before recommending this article for publication, there are some shortcomings for that should be resolve.
General comments
Overall, the study is well designed and presented in a good way, but mostly the literature is not cited. Grammatical and typos must be revised
Abstract
Line 22-23 must be revised. As the knowledge of the medicinal plants is not only limited to Mediterranean region.
Methods results and conclusion must be in sequence in abstract section.
Introduction
Line 48-49 must be cited with relevant studies Pak. J. Bot., 54(3): DOI: http://dx.doi.org/10.30848/PJB2022-3(19),
DOI: 10.56042/ijtk.v21i3.31454,
Introduction is not satisfactory, the author first started introduction from medicinal plants and later discussing about the agriculture.
The authors should discuss about antimicrobial and insecticidal activities of the plants. It need in agriculture as well. It can make a coherence and balanced introduction.
Also discuss harmful effects of the selected insects in this study.
Finally discuss aims and objectives of the study.
Results
Section 2.1
Results showed yield of the EO in specific concentration but what about the other compounds. The ratios must be specified.
Results section should be result and discussion
Conclusion
Add future recommendations as well.
Author Response
Dear Editor, Dear Reviewer,
Please find, in the resubmission section, our final response to the comments received from the three reviewers to paper titled: Phytochemical profile and insecticidal potential of Thymus zygis L., Salvia officinalis L. and Mentha suaveolens Ehrh. essential oils against Ceratitis capitata and Tuta absoluta.
First of all, we would like to thank the editor and the reviewers for their thoughtful comments and efforts towards improving our manuscript by taking time in reading and suggesting modifications to the paper. We highly appreciate the comments received, as they have pointed out a number of issues to be addressed, which have been very useful to amend the paper .
Thank you very much for your kind consideration of this resubmitted version of our manuscript.
Sincerely yours,
Pr. Addi Mohamed and the co-aurhors
Thank you for your very careful review of our paper. A major revision of the paper has been carried out taking into consideration all comments, corrections that you suggested.
Comment 1: Overall, the study is well designed and presented in a good way, but mostly the literature is not cited. Grammatical and typos must be revised.
Response: The literature was updated. We have revised the typos, grammatical errors and the English language editing of the manuscript was revised by Pr Ismail BENBIHI who is a professor of English.
Comment 2: Line 22-23 must be revised. As the knowledge of the medicinal plants is not only limited to Mediterranean region.
Response: This part of the abstract was removed
Comment 3: Methods results and conclusion must be in sequence in abstract section.
Response: The suggested sequence was applied.
Comment 4: Line 48-49 must be cited with relevant studies Pak. J. Bot., 54(3): DOI: http://dx.doi.org/10.30848/PJB2022-3(19)
Response: The following reference was added in manuscript : “Zaman, W., J. Ye, M. Ahmad, S. Saqib, Z.K. Shinwari and Z. Chen. 2022. Phylogenetic exploration of traditional Chinese medicinal plants: a case study on Lamiaceae. Pak. J. Bot., 54(3): DOI: http://dx.doi.org/10.30848/PJB2022-3(19) “ and “ Zaman, W. Ahmad, M. ; Zafar, M. ; Amina, H. ; Lubna ; Saqib, S. ; Ullah, F. ; Ayaz, A. ; Bahadur, S. ; Park, S. Diversity of medicinal plants used as male contraceptives: An initiative towards herbal contraceptives. Indian Journal of Traditional Knowledge Vol 21(3), July 2022, pp 616-624. http://dx.doi.org/10.56042/ijtk.v21i3.31454 “
Comment 5: Introduction is not satisfactory, the author first started introduction from medicinal plants and later discussing about the agriculture.
Response: The first part of introduction related to medicinal and aromatic plants was removed from the manuscript.
Comment 6: The authors should discuss about antimicrobial and insecticidal activities of the plants. It need in agriculture as well. It can make a coherence and balanced introduction.
Response: The insecticidal effect of the studied plants was discussed in the introduction section.
Comment 7: Also discuss harmful effects of the selected insects in this study.
Response: The importance of the 2 insects as a keep pests of citrus and tomato was discussed in the introduction section.
Comment 8: Finally discuss aims and objectives of the study.
Response: The last paragraph of the introduction section mentions the aims of the work.
Comment 9: Results showed yield of the EO in specific concentration but what about the other compounds. The ratios must be specified
Response: The other compounds were not considered during the extraction the EO. The obtained values of yields were expressed as a rate (%) by V/W
Comment 10: Results section should be result and discussion
Response: We have replaced “results” By “results and discussion” in the manuscript.
Comment 11: Add future recommendations as well.
Response: The following sentence was inserted in abstract and conclusion sections : It is considered that the current findings could be a contribution for a development of biopesticides plants-based to be adopted as a component of integrated pest management strategies in citrus and tomato crops.
Round 2
Reviewer 1 Report
The manuscript could be accepted in the present form
Reviewer 2 Report
The authors responded positively with most of the raised points
it could be accepted in the present form